# A Comprehensive Analysis of Physiologic and Hormone Basis for the Difference in Room-Temperature Storability between ‘Shixia’ and ‘Luosanmu’ Longan Fruits

**DOI:** 10.3390/plants11192503

**Published:** 2022-09-25

**Authors:** Libing Long, Tingting Lai, Dongmei Han, Xiaolan Lin, Jianhang Xu, Difa Zhu, Xiaomeng Guo, Yuqiong Lin, Fengyi Pan, Yihang Wang, Ziying Lai, Xinxin Du, Di Fang, Liang Shuai, Zhenxian Wu, Tao Luo

**Affiliations:** 1College of Horticulture, South China Agricultural University/Guangdong Provincial Key Laboratory of Postharvest Science of Fruits and Vegetables/Engineering Research Center of Southern Horticultural Products Preservation, Ministry of Education, Guangzhou 510642, China; 2Institute of Fruit Tree Research, Guangdong Academy of Agricultural Sciences/Key Laboratory of South Subtropical Fruit Biology and Genetic Resource Utilization, Ministry of Agriculture, Guangzhou 510640, China; 3College of Food and Biological Engineering/Institute of Food Science and Engineering Technology, Hezhou University, Hezhou 542899, China

**Keywords:** longan (*Dimocarpus longan* Lour.) fruit, pericarp browning, aril breakdown, ROS scavenging system, abscisic acid (ABA), methyl jasmonate (MeJA), salicylic acid (SA)

## Abstract

Although the effects of phytohormones (mainly salicylic acid) on the storability of longan fruit have been reported, the relationship between postharvest hormone variation and signal transduction and storability remains unexplored. The basis of physiology, biochemistry, hormone content and signalling for the storability difference at room-temperature between ‘Shixia’ and ‘Luosanmu’ longan fruit were examined. ‘Luosanmu’ longan exhibited faster pericarp browning, aril breakdown and rotting during storage. ‘Luosanmu’ pericarp exhibited higher malondialdehyde but faster decreased total phenolics, flavonoid, glutathione, vitamin C, catalase activity and gene expression. Higher H_2_O_2_ and malondialdehyde but lower glutathione, glutathione-reductase and peroxidase activities, while higher activities and gene expressions of polygalacturonase, β-galactosidase and cellulose, lower covalent-soluble pectin, cellulose and hemicellulose but higher water-soluble pectin were observed in ‘Luosanmu’ aril. Lower abscisic acid and methyl jasmonate but higher expressions of *LOX2*, *JAZ* and *NPR1* in pericarp, while higher abscisic acid, methyl jasmonate and salicylic acid together with higher expressions of *ABF*, *JAZ*, *NPR1* and *PR-1* in ‘Luosanmu’ aril were observed. In conclusion, the imbalance between the accumulation and scavenging of active oxygen in ‘Luosanmu’ longan might induce faster lipid peroxidation and senescence-related hormone signalling and further the polymerization of phenolics in pericarp and polysaccharide degradation in aril.

## 1. Introduction

Longan *(Dimocarpus longan* Lour.) is a tropical and sub-tropical fruit belonging to the Sapindaceae family, whose production is mainly distributed in southern Asia including China, Thailand, Vietnam and India [1]. Longan fruit is attractive to customers due to its abundant primary metabolites (such as sugars, organic acids and amino acids) and secondary metabolites (such as phenolic acids, flavonoids, alkaloids and nucleotides) in its edible aril (the hypertrophic tissue attached around the seed coat of longan) [2]. However, the quality deterioration of longan fruit occurs rapidly within a few days under normal temperature storage, mainly manifested as pericarp browning, aril breakdown and decay. The physiological basis of pericarp browning and aril breakdown of longan fruit has been studied extensively in the past few decades [3]. Postharvest pericarp browning of longan fruit was believed to be caused by the oxidation of phenolics by peroxidase (POD), polyphenol oxidase (PPO) [4] and the recently reported Laccase 14-4 [1]. This oxidation process might subsequently be exacerbated by the burst and accumulation of reactive oxygen species [5], peroxidation of membrane lipid [6], and hydrogen peroxide-induced energy deficiency [7]. The above-mentioned physiological disorder might be related to the absence of enzymatic and nonenzymatic ability for scavenging free radicals. In addition to being induced by reactive oxygen species, postharvest longan pericarp browning is also promoted by the destroying of the cellular compartmentation and integrity of the cell membrane, which usually results from water loss [8], pathogen infection [9], and other stresses.

A series of strategies, including chemical, physical, and biological treatments, have been developed to prevent longan fruit from postharvest loss [10,11]. Fumigation with SO_2_ (or other sulphur treatments), traditional fungicide and other useful treatments (such as acid dipping, [12]) are still widely applied in cold-chain logistics and long-period storage of longan fruit due to their powerful inhibition of rotting and browning [10]. However, the use of sulphur-fumigation technology was limited by strict sulfite residue limits and hazards to the environment and human health. In recent years, chemical inhibitors of pericarp browning (propyl gallate, [13]; 2-butanol, [14]; α-aminoisobutyric acid or β-aminoisobutyric acid, [15]), chlorine dioxide (or sodium chlorite as donor) [16,17], coatings (chitosan, [18]), sealing films [19], controlled atmospheres [20], pure oxygen [21], ozone incorporated with some organic acids [22], and NO (sodium nitroprusside used as donor [4]) have shown to be effective controls of longan postharvest loss. Plant hormones act as important factors regulating senescence and responding to biotic and abiotic stresses. Among them, salicylic acid (SA) treatment was reported to suppresses *Phomopsis longanae* Chi-induced disease development of postharvest longan fruit [9]. However, their behaviour, roles and signal transduction in the postharvest deterioration of longan fruit have been rarely investigated.

In previous studies, it was shown that faster development of aril breakdown and worse storability was observed in ‘Fuyan’ longan fruit when compared to ‘Dongbi’ longan fruit stored at room-temperature (25 ± 1 °C) [23,24]. A similar result was found in our previous study in which the difference in low-temperature (4 °C) storability among 14 longan cultivars was compared. It was also shown that postharvest deterioration including mass loss, pericarp browning and aril breakdown varied among different cultivars. More importantly, ‘Shixia’ longan fruit and ‘Luosanmu’ longan fruit showed the best storability and the worst storability, respectively [25]. However, the reason for the great difference in storability between ‘Shixia’ and ‘Luosanmu’ longan fruit remains to be studied. On the other hand, although the effect of plant hormones on the storability of longan (SA: [9,26,27]) and litchi fruit (abscisic acid, ABA [28]; gibberellic acid, GA_3_ [28]; melatonin [29]; methyl jasmonate, MeJA [30]) have been reported increasingly in recent years, their roles in the postharvest storage of longan fruit and their possible relationship with the storability differences among cultivars have not been studied. Herein, in order to uncover the physiologic and hormone basis for the difference in room-temperature storability between ‘Shixia’ and ‘Luosanmu’ longan fruits, we systematically compared the biochemistry, enzyme activities and expression of key genes related to pericarp browning and aril breakdown in ‘Shixia’ to those in ‘Luosanmu’ longan. The contents of ABA, MeJA and SA as well as the expression of genes related to their biosynthesis and signal transduction were also detected in the pericarp and aril of ‘Shixia’ and ‘Luosanmu’ during room-temperature storage. Through a correlation analysis, we screened out the key gene, enzyme and phytohormone, which were highly associated with pericarp browning and aril breakdown. These results are expected to provide new insights into the possible mechanism regulating storability differences among cultivars of longan.

## 2. Results and Discussion

### 2.1. Difference of Quality Deterioration between ‘Shixia’ and ‘Luosanmu’ Longan during the Room-Temperature Storage

The results of chromatic analysis showed that higher *L** (brightness) was found on ‘Luosanmu’ longan pericarp at 0 days after storage (DAS) to 6 DAS and higher *a** value (more red) was observed on ‘Luosanmu’ longan pericarp at 0 DAS to 2 DAS (Figure 1A,B). No significant difference in *b** value (blue to yellow) was found between ‘Shixia’ and ‘Luosanmu’ longan pericarp at harvest (Figure 1C). However, lower *L** and *a** value at the last two days (6 to 8 DAS) as well as lower *b** at 2 DAS to 8 DAS were observed on ‘Luosanmu’ longan pericarp. These results indicated a faster deterioration of appearance in the ‘Luosanmu’ pericarp (Figure 1A–C). The browning of inner pericarp showed a consistent trend with the deterioration of appearance (Figure 1D and Appendix A). More importantly, faster development of aril breakdown and rotting of ‘Luosanmu’ longan fruit was observed (Figure 1E,F and Appendix A). Lower total soluble solid (TSS) content was steadily found in ‘Luosanmu’ longan fruit and lower titratable acid (TA) was detected in ‘Luosanmu’ longan at 0 and 4 DAS (Figure 1G,H).

In our previous results, it was shown that the ‘Shixia’ fruit had the highest storability among the fruits of 14 cultivars, while ‘Luosanmu’ fruit had a poor performance: higher mass loss rate at 25 °C, faster decrease in soluble solid, quicker development of pericarp browning and aril breakdown at 4 °C [25]. In this work, the above results indicated that the quality of ‘Luosanmu’ longan fruit deteriorated faster than ‘Shixia’ longan fruit during room temperature storage (Figure 1; Appendix A). These results were in accordance with the previous report [25].

### 2.2. The Difference of Non-Enzymatic and Enzymatic Radical-Scavenging Capacity in Pericarp between ‘Shixia’ and ‘Luosanmu’ Longan during the Room-Temperature Storage

Interestingly, lower H_2_O_2_ content was observed in ‘Luosanmu’ longan pericarp during the first 6 days of storage (Figure 2A). However, the malondialdehyde (MDA) content in ‘Luosanmu’ longan pericarp was higher during storage, especially at 6 and 8 DAS (Figure 2B). Unexpectedly, lower PPO activity was observed in ‘Luosanmu’ longan pericarp at 4 DAS to 8 DAS (Figure 2C).

Higher POD activity was observed in ‘Luosanmu’ longan pericarp at almost all of the storage stages (Figure 2D). More importantly, the total phenolics and total flavonoid content were rapidly decreased in ‘Luosanmu’ longan pericarp at 4 DAS to 8 DAS and lower than those in ‘Shixia’ longan pericarp (Figure 2E,F). Similarly, lower GSH (reduced glutathione) and vitamin C (VC) were observed in ‘Luosanmu’ longan pericarp at 4 DAS to 8 DAS (Figure 2G,H). The catalase (CAT) activity and ascorbate peroxidase (APX) activity were decreased in both of the ‘Luosanmu’ longan pericarp and ‘Shixia’ longan pericarp throughout the whole storage (Figure 2I,J). Lower CAT activity was only observed in ‘Luosanmu’ longan pericarp at 2 DAS and 8 DAS (Figure 2I). However, significant higher glutathione reductase (GR) activity was detected in ‘Luosanmu’ longan pericarp throughout the whole storage (Figure 2K).

The balance between the accumulation of active oxygen and its (enzymatic and non-enzymatic) scavenging capacity was an important factor affecting the browning and deterioration of longan pericarp [5,6,7]. In this work, decreased contents of endogenous antioxidant substances (VC and GSH) and increased content of MDA indicated a decreased capacity for active oxygen scavenging and accelerated membrane lipid peroxidation in ‘Luosanmu’ longan pericarp. Although lower H_2_O_2_ but higher POD activity was observed in ‘Luosanmu’ pericarp, their roles may be explained in another way. Apart from its role in active oxygen metabolism, it was recently reported that POD but not laccases act as an essential factor for lignification in the Casparian strip of Arabidopsis [31]. Thus, higher POD activity might contribute to mediate the participation of H_2_O_2_ in lignin polymerization in the ‘Luosanmu’ longan pericarp. Although the decrease in total phenolic content (TPC) and total flavonoid content (TFC) was consistent with this hypothesis, further evidence, such as histochemical results, are necessary. In total, the above-mentioned results indicated that the more intense physiological and biochemical deterioration in ‘Luosanmu’ pericarp led to a faster disrupted cellular membrane structure and loss of cellular compartmentalization, resulting in turn in the full contact of PPO and POD with phenolic substrates. Subsequently, quicker phenol oxidation and browning may also occur in ‘Luosanmu’ pericarp [5,31].

### 2.3. The Difference of Non-Enzymatic and Enzymatic Radical-Scavenging Capacity in Aril between ‘Shixia’ and ‘Luosanmu’ Longan during the Room-Temperature Storage

Higher H_2_O_2_ and MDA content were observed in ‘Luosanmu’ longan aril in the later storage stages (4 to 8 DAS) (Figure 3A,B). Unexpectedly, higher total phenolic content was observed in ‘Luosanmu’ longan aril throughout the whole storage period (Figure 3C). Higher total flavonoid content at 0, 6 and 8 DAS but lower total flavonoid content at 2 and 4 DAS was detected in ‘Luosanmu’ longan aril (Figure 3D). The mainly endogenous antioxidants GSH showed a lower level in ‘Luosanmu’ longan aril at 2 DAS and 8 DAS (Figure 3E) but VC showed higher accumulation in ‘Luosanmu’ longan aril throughout the whole storage period (Figure 3F). Lower POD activity (Figure 3G) and GR activity (Figure 3J) but higher APX activity (Figure 3I) were found in ‘Luosanmu’ longan aril throughout the room-temperature storage period. Higher CAT activity in ‘Luosanmu’ longan aril was only observed at 4 DAS (Figure 3H).

The burst and accumulation of active oxygen is an important factor inducing aril breakdown of longan fruit. In the previous research, it was reported that treatment with hydrogen peroxide can lead to an increase in gene expression and enzyme activities related to cell wall polysaccharide degradation, which can boost the decomposition of cell wall polysaccharide and thus result in the accelerated softening and the expedited breakdown of postharvest longan pulp [32]. The results in the current work indicated that the accelerated lipid peroxidation and higher H_2_O_2_ accumulation together with the lower enzymatic scavenging capacity of free radical throughout the storage might result in faster aril breakdown in ‘Luosanmu’ longan (Figure 1E, Figure 3 and S1).

### 2.4. The Difference of Pectin and Cellulose Components and Activities of Hydrolase Enzymes in Aril between ‘Shixia’ and ‘Luosanmu’ Longan during the Room-Temperature Storage

The covalent-soluble pectin content at 4 DAS to 8 DAS was lower in ‘Luosanmu’ longan aril, which rapidly decreased, while that in ‘Shixia’ longan aril increased in the later storage stages (Figure 4A). Lower ionic-soluble pectin content was detected in ‘Luosanmu’ longan aril only at 8 DAS (Figure 4B), but higher water-soluble pectin content was found in ‘Luosanmu’ longan aril throughout storage (except 4 DAS) (Figure 4C). In accordance with the above results, higher polygalacturonase (PG) and β-galactosidase (β-Gal) activity was also found in ‘Luosanmu’ longan aril throughout the storage period (Figure 4E,F). However, no significant difference in pectinesterase (PE) activity was observed between ‘Luosanmu’ and ‘Shixia’ longan aril (Figure 4D). In addition, lower cellulose content at 4 DAS and 6 DAS (Figure 4G), but lower hemicellulose content at 6 DAS and 8 DAS (Figure 4H), was detected in ‘Luosanmu’ longan aril. It was interesting that higher cellulose (CX) activity was also observed in ‘Luosanmu’ longan aril throughout the entire storage period (Figure 4I).

Pectin and cellulose are the main components of the plant’s cell wall. Softening and deterioration of longan fruit texture usually involves the degradation of pectin and cellulose [32]. The above results indicated the more intense hydrolysis of pectin and cellulose in ‘Luosanmu’ longan aril, which might result in a faster aril breakdown. The quicker cell wall polysaccharide degradation in ‘Luosanmu’ might be related to its endogenously higher accumulation of hydrogen peroxide (Figure 3A). These results are in accordance with previously reported studies in which exogenous reactive oxygen species (ROS) induce the aril breakdown of postharvest longan [32].

### 2.5. Difference of ABA, MeJA, SA Content and the Expression of Genes Related to Radical- Scavenging and Hormones between ‘Shixia’ and ‘Luosanmu’ Fruit during the Room-Temperature Storage

The analysis of hormone contents showed that the ABA content in both of ‘Luosanmu’ and ‘Shixia’ pericarp increased slowly and reached the highest level at 6 DAS and then decreased. However, ABA content in the ‘Luosanmu’ pericarp was consistently lower than that in the ‘Shixia’ pericarp (Figure 5A). The MeJA content in the ‘Shixia’ pericarp firstly decreased but then raised up, while the MeJA content in the ‘Luosanmu’ pericarp slowly declined and was significantly lower than that in the ‘Shixia’ pericarp at 6 DAS and 8 DAS (Figure 5B). Significantly lower SA content at 0 DAS and 2 DAS but higher SA content at 6 DAS was observed in ‘Luosanmu’ pericarp (Figure 5C). However, higher ABA at 8 DAS and higher MeJA content at 2 DAS, 6 DAS and 8 DAS were found in the ‘Luosanmu’ longan aril (Figure 5D,E). Similarly, higher SA content at 0 DAS and 6 DAS was detected in the ‘Luosanmu’ longan aril (Figure 5F). In total, compared to ‘Shixia’ longan, in the ‘Luosanmu’ longan pericarp, the content of ABA and MeJA was totally lower, while the SA content was lower at the earlier storage stages but higher at the later storage stages; in the ‘Luosanmu’ longan aril, the content of ABA, MeJA and SA was consistently higher at 6 DAS and/or 8 DAS.

The expressions of the key genes involved in ABA, MeJA and SA synthesis and signal transduction as well as the genes related to browning, oxygen radical scavenging and disease response were analyzed in longan pericarp (Figure 5G). The expression of the genes related to ABA synthesis (*NCED1*, nine-*cis*-epoxycarotenoid dioxygenase; *ABA2*, *xanthoxin dehydrogenase*; *AAO3*, *abscisic aldehyde oxidase 3*) decreased in ‘Shixia’ pericarp but up-regulated in the ‘Luosanmu’ pericarp at the later stages (6 to 8 DAS). It was worthy to note that the genes related to MeJA synthesis (*LOX2*, lipoxygenase 2) and SA synthesis (*PAL1*, *phenylalnine ammonialyase 1*), as well as the genes for hormone signal transduction (*ABF*, *ABA responsive element binding factor*; *JAZ, jasmonate ZIM domain- containing protein*; *NPR1, non-expressor of pathogenesis-related gene 1*) showed higher up-regulation in the ‘Luosanmu’ pericarp than those in the ‘Shixia’ pericarp during storage. On the other hand, significantly up-regulated expression of *PPO* and *PR-1* (*pathogenesis-related protein 1*) was detected in both of the ‘Luosanmu’ pericarp and ‘Shixia’ pericarp throughout storage, but higher expression of the genes related to oxygen radical-scavenging at the later stages (*GR*, *GPX-glutathione peroxidase* and *POD* at 4 to 8 DAS; *CAT* and *APX* at 8 DAS) was observed in the ‘Luosanmu’ pericarp. 

The expressions of the key genes involved in ABA, MeJA and SA synthesis and signal transduction as well as the genes related to aril breakdown, oxygen radical scavenging and disease response were also analyzed in longan aril (Figure 5H). It was interesting to note that the expressions of genes for signal transduction of ABA (*ABF*), JA (*JAZ*) and SA (*NPR1* and *PR-1*) were significantly up-regulated in the ‘Luosanmu’ aril when compared to those in the ‘Shixia’ aril. Moreover, higher expression of genes related to ABA synthesis (*NCED*, *ABA2* and *AAO3*) and genes for MeJA synthesis (*LOX2*) at the later stages (6 DAS and 8 DAS) but higher expression of *PAL1* (for SA synthesis) throughout storage were observed in the ‘Luosanmu’ aril (Figure 5H). Similarly, higher expression of genes related to oxygen radical-scavenging (*APX*, *GPX*, and *POD*) at the later stages (6 DAS and 8 DAS) or throughout storage (*CAT* and *GR*) was observed in the ‘Luosanmu’ aril (Figure 5H). More importantly, a significantly higher expression of genes related to pectin degradation (*PG*, *PE*) throughout the storage period and higher expression of genes for cellulose decomposition (*CX* and *β-Gal*) at 6 DAS and 8 DAS were found in the ‘Luosanmu’ aril (Figure 5H). 

ABA and ETH are usually considered as the two most important regulatory factors of maturity and senescence in climacteric and non-climacteric fruits [33,34]. Especially, ABA has been proven to play a vital role in the ripening and postharvest senescence of non-climacteric fruits such as grape [35], blueberry (*Vaccinium corymbosum* L.) [36], strawberry [37], and citrus [33]. As an important regulator of the defence system, JA shows various defence responses from pathogen to environment stresses [34]. As another inevitable member of the defence system, SA usually enhances environmental resistance at a low concentration, but it may induce cell necrocytosis at a high concentration [34]. The results in this work indicated that the different fluctuations of these three hormones between ‘Shixia’ and ‘Luosanmu’ longan may be the reason for the differential postharvest deterioration between these two cultivars. More importantly, the more intense ABA, MeJA and SA signal transduction and up-regulated expression of oxygen radical-scavenging genes in the ‘Luosanmu’ pericarp might be a response and resistance to biotic and abiotic stress during quality deterioration. Additionally, the more intense ABA, MeJA and SA signal transduction and up-regulated expression of oxygen radical-scavenging genes in ‘Luosanmu’ aril might be related to aril breakdown, which is regulated at the metabolic (Figure 4A–C,G,H, degradation of pectin and cellulose), enzymatic (Figure 4E,F,I) and transcriptional level (Figure 5H). It was interesting to note that ABA, MeJA and SA response elements were the main cis-elements related to hormones in the promoter sequences of our tested genes (used for qRT-PCR in this work, see Appendix A). Therefore, it was reasonable to speculate that ABA, JA and SA signals were involved in the regulation of reactive oxygen scavenging, metabolism of antioxidants and polysaccharide degradation during the development of aril breakdown and pericarp browning.

### 2.6. The Correlation of Physiological Parameters, Enzyme Activities and Gene Expression with the Longan Deterioration

The Pearson correlation of physiological parameters, enzyme activities and expression of the selected genes with longan deterioration containing pericarp browning, rotting rate and aril breakdown was analyzed (Figure 6). The results indicated that the pericarp browning index showed an extremely significant positive correlation with the expression of *LOX2* (r = 0.95, *p*-value = 2.58 × 10^−5^) and rotting rate (r = 0.95, *p*-value = 3.79 × 10^−5^) and a high positive correlation with the MDA content (r = 0.92, *p*-value = 1.58 × 10^−4^), GR activity (r = 0.91, *p*-value = 2.51 × 10^−4^), the expression of *PAL1* (r = 0.85, *p*-value = 1.80 × 10^−3^), *JAZ* (r = 0.84, *p*-value = 2.47 × 10^−3^) and *PR-1* (r = 0.81, *p*-value = 4.23 × 10^−3^). Moreover, the pericarp browning index showed a moderate positive correlation with the expression of *NPR1* (r = 0.77, *p*-value = 8.70 × 10^−3^), *GR* (r = 0.73, *p*-value = 1.69 × 10^−2^), *PPO* (r = 0.68, *p*-value = 3.01 × 10^−2^), *APX* (r = 0.65, *p*-value = 4.18 × 10^−2^), and *GPX* (r = 0.65, *p*-value = 4.21 × 10^−2^), POD activity (r = 0.71, *p*-value = 2.20 × 10^−2^) and SA content (r = 0.63, *p*-value = 5.06 × 10^−2^) (Figure 6A). However, the pericarp browning index showed a high negative correlation with the *b** (r = −0.91, *p*-value = 2.22 × 10^−4^) and TFC (r = −0.81, *p*-value = 4.41 × 10^−3^), and showed a moderate negative correlation with the TPC (r = −0.74, *p*-value = 1.44 × 10^−2^), CAT activity (r = −0.71, *p*-value = 2.15 × 10^−2^) and GSH content (r = −0.69, *p*-value = 2.75 × 10^−2^) (Figure 6A). It is worth noting that the pericarp browning index unexpectedly showed a high negative correlation with the PPO activity (r = −0.90, *p*-value = 3.88 × 10^−4^) (Figure 6A). 

The aril breakdown index showed an extremely significant positive correlation with the expression of *NPR1* (r = 0.98, *p*-value = 4.27 × 10^−7^) and *JAZ* (r = 0.97, *p*-value = 3.29 × 10^−6^), the rotting rate (r = 0.98, *p*-value = 7.40 × 10^−7^) and H_2_O_2_ content (r = 0.97, *p*-value = 4.25 × 10^−6^). The aril breakdown index also showed a high positive correlation with the expression of β-Gal (r = 0.94, *p*-value = 4.14 × 10^−5^) and ABF (r = 0.86, *p*-value = 1.50 × 10^−3^), TPC (r = 0.88, *p*-value = 8.05 × 10^−4^), ABA content (r = 0.85, *p*-value = 1.78 × 10^−3^) and MDA content (r = 0.80, *p*-value = 5.00 × 10^−3^). The aril breakdown index only showed a moderate positive correlation with the expression of *PE* (r = 0.78, *p*-value = 7.55 × 10^−3^) and *CX* (r = 0.64, *p*-value = 4.83 × 10^−2^), water-soluble pectin content (r = 0.76, *p*-value = 1.02 × 10^−2^), TFC (r = 0.74, *p*-value = 1.44 × 10^−2^), APX activity (r = 0.71, *p*-value = 2.17 × 10^−2^), MeJA content (r = 0.65, *p*-value = 4.33 × 10^−2^) and VC content (r = 0.64, *p*-value = 4.78 × 10^−2^). In addition, the aril breakdown index was negatively correlated with the hemicellulose content (r = −0.82, *p*-value = 3.76 × 10^−3^), covalent-soluble pectin content (r = −0.75, *p*-value = 1.34 × 10^−2^), and ionic-soluble pectin content (r = −0.74, *p*-value = 1.41 × 10^−2^). 

In the above-mentioned results, due to their strong correlation with the pericarp browning index, it was suggested that the expression of MeJA synthesis-related gene *LOX2*, SA synthesis-related gene *PAL1*, as well as the genes related to MeJA and SA signal transduction (*JAZ*, *PR-1* and *NPR1*) might be used as indicator parameters of pericarp browning. In the previous study, SA treatment (0.3 g·L^−1^) was reported to suppress *Phomopsis longanae* Chi-induced disease development of postharvest longan fruit [9]. It was known that SA usually enhances environmental resistance at a low concentration, but it may induce cell necrocytosis at a high concentration [34]. In this study, the concentration of SA in longan fruit fluctuated in a low range (pericap, about 6 to 11 ng·g^−^^1^ FW; aril, about 15 to 22 ng·g^−^^1^ FW). Therefore, the fluctuation of SA might be a response to postharvest environmental stress. On the other hand, the above-mentioned results indicated that the faster degradation of pectin and cellulose at the later storage stages in ‘Luosanmu’ longan aril might be related to the up-regulated expression of genes of MeJA (*JAZ*), SA (*NPR1*) and ABA (*ABF*) signal transduction, ABA content, SA content and H_2_O_2_ accumulation. However, evidence for the direct regulation of longan fruit pericarp browning and aril breakdown by ABA, SA and MeJA is still lacked. Future research is expected to involve identifying the roles of ABA, MeJA and SA in regulating longan pericarp browning and aril breakdown through molecular interaction techniques such as yeast one hybrid (Y1H), electrophoretic mobility shift assay (EMSA) and luciferase reporter assay.

## 3. Materials and Methods

### 3.1. Preparation, Treatment and Storage of Longan Fruit

The commercial mature longan fruit (cv. Shixia and cv. Luosanmu) used for storage was harvested at about 100 days after flowering in an orchard with standardized production in Guangzhou (Guangdong province, south China). The harvested fruit were collected and immediately transported to the laboratory. More than 500 fruits with no disease and no damage were selected and dipped in 500 mg·L^−1^ prochloraz solution for 2 min. After a natural air-dry at room temperature for 30 min, the fruit were packed into trays (about 20 fruits per tray) with 0.01 mm thick polyethylene film and stored at room temperature for 8 days (RT, 25 ± 1 °C; 85% relative humidity). Photographs recording changes in appearance and aril breakdown of ‘Shixia’ and ‘Luosanmu’ fruit during room-temperature storage are shown in Appendix A.

### 3.2. Determination of Chromatic Value

Ten fruits were randomly selected from one tray, respectively, at 0, 2, 4, 6 and 8 days after storage (DAS). The *L**, *a** and *b** values on the equatorial plane of each fruit were measured using a colour analyzer (NH310, 3nh Technology Co., Ltd., Shenzhen, China). The value of *L** indicates brightness ranging from 0 (black) to 100 (white). The negative value to positive value of *a** indicates green to red, respectively. The negative value to positive value of *b** indicates blue to yellow, respectively. The measurement at each time-point was subjected to three repetitions.

### 3.3. Determination of Pericarp Browning Index, Aril Breakdown Index and Rotting Rate

Three trays of fruits were randomly selected for examining rotting rate at 0, 2, 4, 6 and 8 DAS. In total, 30 fruits (10 fruits per tray) were used to determine the browning index of inner pericarp and the aril breakdown index. Evaluation of browning index and aril breakdown index at each time point was repeated three times.

The browning index of inner pericarp was calculated according to a reported method [11]. The browned area (BNA) of each fruit was estimated using the following scales: score = 0, no browning; score = 1, BNA was 0.1% to 25%; score = 2, BNA was 25.1% to 50%; score = 3, BNA was 50.1% to 75%; score = 4, BNA was 75.1% to 99.9%; score = 5, BNA was 100%. The pericarp browning index was calculated according to Equation (1):Pericarp browning index = ∑(BNA score of each fruit)/30(1)

The breakdown area (BDA) of each longan aril was evaluated using the following scales: score = 0, no breakdown; score = 1, BDA was 0.1% to 25%; score = 2, BDA was 25.1% to 50%; score = 3, BDA was 50.1% to 75%; score = 4, BDA was 75.1% to 99.9%; score = 5, BDA was 100%. The aril breakdown index was calculated using Equation (2):Aril breakdown index = ∑(BDA score of each fruit)/30(2)

### 3.4. Sampling and Determination of Total Soluble Solid (TSS), Titratable Acid (TA) and Vitamin C (VC)

Thirty longan fruit, respectively, sampled at 0, 2, 4, 6 and 8 DAS, were separated into pericarp and aril. The collected pericarp of fruit and one half of the collected aril were immediately frozen in liquid nitrogen, ground and stored at −80 °C until used. The other half of collected aril was used for juicing and determination of TSS and TA content by a digital refractometer (PAL-BX/ACID F5, ATAGO Co., Ltd., Tokyo, Japan). Sampling, TSS and TA analysis were performed with three repeats (one bag per repeats). The VC content of longan fruit was determined by an ultraviolet-visible spectrophotometer method with some adjustments [38]. Ten microliter longan juice was diluted with 9900 μL oxalic acid solution (2%, *w*/*v*, pH6.0 adjusted with NaOH) and then the absorbance was measured at 267 nm. The VC content was calculated by a standard curve.

### 3.5. Assay of Total Phenolics, Total Flavonoid, and Reduced Glutathione (GSH)

The total phenolic content (TPC) and total flavonoid content (TFC) of longan pericarp and aril were measured according to the previously reported methods [2]. The ethanolic extract used for determination of TPC and TFC contents was prepared by homogenizing 0.1 g frozen pericarp or 0.3 g frozen aril, which was extracted with 1 mL 80% ethanol using a vigorously vortex and ultrasonication (KQ 5200E, Kunshan Ultrasonic Instrument Co., Ltd., Kunshan, China) for 60 min. The samples were then extracted at 4 °C overnight and subjected to a ultrasonic extraction (aril 1 h, pericarp 2 h, with ice bath). After a centrifugation at 12,000× *g* for 10 min, the supernatant of each sample was used to determine TPC and TFC. The TFC was measured at 510 nm according to the method reported by Lai et al. [2] using a standard curve of rutin and the TPC was assayed at 760 nm by the Folin–Ciocalteu method using gallic acid as the standard [2].

The extract used for determination of GSH content was prepared by homogenizing 0.1 g pericarp powder with 900 µL PBS buffer (50 mM, pH 7.0) or 0.2 g aril powder with 800 µL PBS buffer. After being fully vortexed, the sample was subjected to a centrifugation at 2500× *g* and 4 °C for 10 min [5,6]. The GSH content of longan pericarp and aril was measured based on its reaction with DTNB by using a commercialized GSH assay kit (A006-2-1, Nanjing Jiancheng Bioengineering Institute, Nanjing, China, http://www.njjcbio.com/products.asp?id=1532, accessed on 01 August 2022). The absorbance at 405 nm was measured and the GSH content was calculated according to a standard curve.

### 3.6. Determination of PPO Activity, Malondialdehyde (MDA) Content and H_2_O_2_ Content

PPO activity was determined according to a method reported by Zhang et al. [39] with some modifications. Sample (0.2 g powder) was added into a tube and then fully mixed with 1 mL precooled 50 mM PBS (pH 7.0) and 0.2 g PVP. The sample was extracted by a ultrasonication for 30 min in an ice-bath. After being centrifuged at 12,000 × *g* and 4 °C for 20 min, the supernatant of the sample was removed to a new tube and used as the enzyme solution. The reaction was started by adding 0.2 mL of enzyme solution to 2.8 mL pyrocatechin solution (10 mM). The absorbance at 398 nm within 180 s was continuously recorded. The change in 0.01 OD_398_ nm per minute was recorded as one enzyme activity unit (U). The result was expressed as U·g^−1^ FW (*n* ≧ 3).

The MDA content was determined using the thiobarbituric acid (TBA) method with some adjustments [39]. Sample (0.2 g powder) was fully mixed with 1 mL PBS buffer (pH 7.8, containing 0.2 mM EDTA) and then extracted by ultrasonication for 30 min in an ice-bath. After a centrifugation at 13,000× *g* and 4 °C for 20 min, 1 mL supernatant was mixed with 3 mL 0.6% TBA (dissolve in 10% trichloroacetic acid, TCA) and then heated in boiling water for 30 min. The solution was cooled and centrifuged at 5000× *g* for 10 min. The absorbance of supernatant at 532 nm was recorded and the nonspecific absorption at 600 nm and 450 nm was subtracted. The solution of 4 mL 0.6% TBA (dissolved in 10% TCA) was set as a blank. Each sample was assayed by three repeats.

The H_2_O_2_ content was detected using the method described by Zhang et al. [39] with some modifications. The longan sample (0.1 g powder) was homogenized with 1 mL precooled acetone and shaken for 30 s. Then, the sample was extracted using a ultrasonication for 10 min in an ice-bath. After being centrifuged at 10,000× *g* and 4 °C for 20 min, 0.5 mL supernatant was mixed with 0.05 mL 10% Ti(SO_4_)_2_ (*v*/*v*, dissolved in concentrated HCl) and 0.1 mL of concentrated NH_3_·H_2_O. After a reaction for 5 min, the solution was centrifuged at 10,000× *g* and 4 °C for 15 min. The precipitate was washed three times with precooled acetone until the pigments were removed, and then dissolved in 3 mL H_2_SO_4_ (2 M). After being centrifuged at 5000× *g* and 4 °C for 15 min, the absorbance of the supernatant at 415 nm was detected. The H_2_O_2_ content was measured using a standard curve of H_2_O_2_ and expressed as μmol·g^−1^ FW (*n* ≧ 3).

### 3.7. Measurement of Peroxidase, Catalase, Ascorbate Peroxidase and Glutathione Reductase Activities

The activity of POD was assayed according to a change in absorbance at 470 nm caused by the generation of tetraguaiacol from guaiacol in the presence of H_2_O_2_ [6,39] with some modifications. The longan sample (0.1 g pericarp powder or 0.2 g aril powder) was homogenised with 1 mL PBS buffer (50 mM, pH 5.5, containing 2% PVP, *w*/*v*) and then subjected to extraction by ultrasonication for 10 min in an ice-bath. After the sample was centrifuged at 15,000× *g* and 4 °C for 20 min, the supernatant was used as the enzyme solution. The reaction was started by mixing 30 μL of supernatant with 90 μL PBS buffer, 30 μL 2% H_2_O_2_ and 30 μL 50 mM guaiacol, and then kept at 35 °C for 10 min and stopped by adding 60 μL 20% (*w*/*v*) TCA. A change in 0.01 in the absorbance at 470 nm per minute was recorded as one enzyme unit. The result was expressed as U·g^−1^ FW (*n* ≧ 3).

The longan sample (0.1 g powder) was fully mixed with 1 mL precooled 0.05 M PBS buffer (pH 7.0, containing 1 mM EDTA and 2 mM ascorbic acid) by shaking for 30 s, and then was extracted by a ultrasonication for 10 min in an ice-bath. After being centrifuged at 12,000× *g* and 4 °C for 20 min, the supernatant was used for measuring catalase (CAT) and ascorbate peroxidase (APX) activities [5].

The reaction mixture for assaying CAT activity consisted of 2.9 mL 0.3% H_2_O_2_ and 0.1 mL of enzyme solution. Decomposing of H_2_O_2_ by CAT was monitored by continuously assaying the absorbance at 240 nm. The change in the absorbance by 0.01 per minute was recorded as one unit (U) of CAT activity [5]. The reaction mixture for APX activity determination consisted of 0.3 mL 0.5 mM ascorbic acid, 0.3 mL 1 mM H_2_O_2_ and enzyme solution dissolved with PBS (0.2 mL enzyme solution of aril and 2.2 mL PBS or 0.1 mL enzyme solution of pericarp with 2.3 mL PBS). The change in the absorbance at 290 nm by 0.01 per minute was regarded as one unit (U) of APX activity [5]. The result was expressed as U·g^−1^ FW (*n* ≧ 3).

The activity of glutathione reductase (GR) was measured by a plant GR ELISA kit (Mlbio, Shanghai, China). Longan aril or pericarp was ground into powder by liquid nitrogen, and then 0.1 g powder was homogenised with 1 mL PBS buffer (10 mM, pH 7.0) in an ice bath. The sample was fully vortexed and then extracted by ultrasonication for 20 min. The sample was centrifuged at 8000× *g* and 4 °C for 30 min, and the supernatant was used for determining GR activity. The sample, standards, and HRP labelled detection antibody were added to the micro-plate coated with the antibody (double-antibody one-step sandwich method) and then incubated at 37 °C for 1 h. After being thoroughly washed, TMB solution was added and used as the substrate, which was converted to blue under the catalysis of peroxidase, and finally yellow under the action of acid. Each sample at each time point was subjected to three repeated tests. The absorbance at 450 nm was measured and the GR activity was expressed as U·g^−1^ FW.

### 3.8. Determination of Pectin, Cellulose, and Hemicellulose Contents

The content of the cell wall component (CWC) in longan aril was measured according to the method described by Chen et al. [40] with some modifications. Longan sample (3 g aril) powder was homogenized with 10 mL 80% ethanol and then boiled for 30 min with continuous stirring to destroy the cell wall and inactivate enzymes. After being centrifuged at 7500× *g* and 4 °C for 15 min, the supernatant was removed and the insoluble precipitate was boiled and washed by 10 mL 80% ethanol again. This step was repeated until the sugar in the sample was completely washed out. The Molish reaction (α-naphthol-concentrated H_2_SO_4_ acid) was used to detect whether the sugar was fully removed. After being washed with chloroform: methanol (1:1, *v*/*v*), the residue was dipped in dimethyl sulfoxide (90%, *v*/*v*) and kept for 8 h. The starch in the sample was detected by the I2-KI method. Subsequently, the filtered residue was washed with acetone, dried to a constant weight and used as CWC.

Then, 20 mg CWC powder was dipped in 1.3 mL ultra-pure water for 6 h. After being centrifuged at 12,000× *g* and 4 °C for 10 min, the supernatant was used to detect water-soluble pectin (WSP). The residue was extracted with 1.3 mL 50 mM sodium acetate buffer (pH 6.8, containing 1 mM EDTA) for 6 h. The supernatant was collected after a centrifugation (12,000× *g*, 4 °C, 10 min) and used for analyzing ion-binding pectin (ISP). The residue was extracted with 1.3 mL 50 mM Na_2_CO_3_ (containing 20 mM NaBH_4_) for 6 h and the supernatant was collected after a centrifugation (12,000× *g*, 4 °C, 10 min) to measure covalently bound pectin (CSP). The residue was extracted with 1.3 mL of 50 mM NaOH (containing 100 mM NaBH_4_) for 6 h and the supernatant was collected after a centrifugation (12,000× *g*, 4 °C, 10 min) to quantify hemicellulose. The residue was washed three times with 0.1 M KOH solution and 8 mM Na_2_SO_3_ to remove uronic acid and lignin and then immersed in 5 mL of 8 M KOH for 6 h. After a centrifugation at 7500× *g* and 4 °C for 15 min, the residue was washed with 80% ethanol three times and dried to a constant weight to obtain cellulose.

The supernatant of WSP (ISP or CSP) was mixed with 1 mL distilled water and 1 mL of concentrated H_2_SO_4_, and then boiled for 20 min. The sample was then cooled in an ice-bath, mixed with 40 μL 95% ethanol (containing 7 mM carbazole) and kept in the dark for 30 min. The absorbance of the solution at 530 nm was recorded. The content of WSP (ISP or CSP) was calculated according to a standard curve of galacturonic acid and expressed as mg·g^−1^ FW (*n* ≧ 3).

The solution for hemicellulose (0.5 mL) (or the whole residue for cellulose) was mixed with 2.5 mL of 2 M H_2_SO_4_ and incubated in boiling water for 5 h. After being cooled in an ice-bath, the sample was mixed with 2 M H_2_SO_4_ to be 8 mL. Then, 0.2 mL supernatant was mixed with 50 μL of anthrone solution and 0.5 mL of concentrated H_2_SO_4_. After incubation in boiling water for 5 min, the absorbance of the solution at 620 nm was recorded. The content of hemicellulose or cellulose in each sample was calculated according to a standard curve of glucose and expressed as mg·g^−1^ FW (*n* ≧ 3).

### 3.9. Determination of Pectinesterase, Polygalacturonase, β-Galactosidase and Cellulose Activities

The activities of pectinesterase (PE), polygalacturonase (PG), β-galactosidase (β-Gal) and cellulose (CX) in longan aril was measured by a plant ELISA kit (Mlbio, Shanghai, China). Longan aril was firstly ground into powder by liquid nitrogen, then 0.2 g powder was homogenized with 1 mL PBS buffer (10 mM) in a tube (on ice bath). The sample was fully vortexed and extracted by a ultrasonication for 20 min. After being centrifuged at 8000× *g* and 4 °C for 30 min, the supernatant was used for the assay of PE. Other supernatant samples were prepared according to the above-mentioned procedure for determining PG, β-Gal and CX activity, respectively. Each sample at each time point was subjected to three repeated tests. The supernatant, standard solutions, and HRP labeled detection antibody were added to the microwell-plate coated with the antibody (a double-antibody one-step sandwich method). After being incubated at 37 °C for 1 h, the reaction solution in each well was thoroughly washed. TMB solution used as substrate was added and converted to blue under the catalysis of peroxidase, and finally yellow under the action of acid. The absorbance at 450 nm was measured and the activities of PE, PG, β-Gal and CX were calculated and expressed as U·g^−1^ FW.

### 3.10. Determination of Abscisic Acid (ABA), Methyl Jasmonate (MeJA) and Salicylic Acid (SA) Content

The contents of ABA, MeJA and SA in longan pericarp and aril were measured by plant ELISA kits (Mlbio, Shanghai, China) [41]. Pericarp and aril samples were ground into powder by liquid nitrogen, and then 0.2 g of powder was homogenized with 1 mL PBS buffer (10 mM) in a test tube with an ice bath. After being fully vortexed, the sample was extracted using a ultrasonication for 20 min and then centrifuged at 8000× *g* and 4 °C for 30 min. The supernatant was used for quantification of ABA, MeJA and SA. The sample supernatant, standard solutions, and HRP labeled detection antibody were added to the microwell-plate coated with the antibody (a double-antibody one-step sandwich method). After being incubated at 37 °C for 1 h, the reaction solution in each well was thoroughly washed. TMB solution used as the substrate was added and converted to blue under the catalysis of peroxidase, and finally yellow under the action of acid. Then, the absorbance in each well was measured at 450 nm and the contents of ABA, MeJA and SA were calculated and expressed as ng·g^−1^ FW.

### 3.11. RNA Isolation and qRT-PCR Analysis

The TransZol Plant RNA extraction kit (TransGen Biotech Co., Ltd., Beijing, China) was used for RNA extraction of longan aril, and Quick RNA Isolation Kit (Huayueyang Biotechnology (Beijing) Co., Ltd., Beijing, China) was used for RNA extraction of longan pericarp. The cDNA was synthesized using Evo M-MLV RT MIX kit (containing one-step gDNA Eraser) for qPCR kit (Yeasen Biotechnology (Shanghai) Co., Ltd., Shanghai, China). Primers of 15 genes for qRT-PCR analysis of longan pericarp and 18 genes for qRT-PCR analysis of longan aril were designed online by Primer 5.0 and listed in Appendix A. The actin gene was used as a reference gene to normalize expression levels of the target genes across different samples. The designed primers were synthesized by Sangon Biotech (Shanghai) Co., Ltd. (Shanghai, China). Hifair^®^ III one step RT-qPCR SYBR Green Kit 11143ES kit (Yeasen Biotechnology (Shanghai) Co., Ltd., Shanghai, China) was used for qRT-PCR, which was carried out in a Bio-Rad CFX-384 Touch Real-Time PCR Detection System (Bio-Rad Laboratories (Shanghai) Co., Ltd., Shanghai, China). The relative expression was calculated using the 2^−ΔΔ^Ct method [42].

### 3.12. Statistical Analysis

The results obtained were expressed as the mean ± SD (*n* ≧ 3). The statistical significance between the control and the treated sample was analyzed by Paired-samples *t*-tests. Additionally, correlation coefficients were analyzed by SPSS software package release 17.0 (SPSS Inc. Chicago, IL, USA).

## 4. Conclusions

The pericarp browning, aril breakdown and rotting of ‘Luosanmu’ longan showed a faster development than those of ‘Shixia’ longan during room-temperature storage. The ‘Luosanmu’ pericarp exhibited a higher malondialdehyde content and pericarp browning index, which might be related to the weaker scavenging capacity of active oxygen: faster decreased antioxidants (TPC, TFC, glutathione and VC) and lower CAT activity and gene expression. Lower ABA, MeJA, and accumulation of H_2_O_2_ but higher expression of *LOX2*, *JAZ* and *NPR1* indicated different hormone signalling between ‘Shixia’ and ‘Luosanmu’ in the deterioration of pericarp. In line with the faster development of aril breakdown, weaker scavenging capacity of active oxygen (higher H_2_O_2_ and MDA content but lower glutathione content, GR and POD activity) and more intense and rapid hydrolysis of cellulose and pectin (higher activities and gene expressions of *PE*, *PG*, *β-Gal* and *CX*; lower covalent-soluble pectin, cellulose and hemicellulose content but higher water-soluble pectin content) were observed in ‘Luosanmu’ aril. Higher content of ABA, MeJA and SA together with the higher expressions of *ABF*, *JAZ*, *NPR1* and *PR-1* at the later storage stages indicated the important roles of these three hormone signals in the postharvest senescence of ‘Luosanmu’ longan aril. To sum up, the enhancement of enzymatic and non-enzymatic free radical scavenging abilities and the application of appropriate exogenous hormone treatment are expected to delay the postharvest quality deterioration of longan fruit.

## Figures and Tables

**Figure 1 plants-11-02503-f001:**
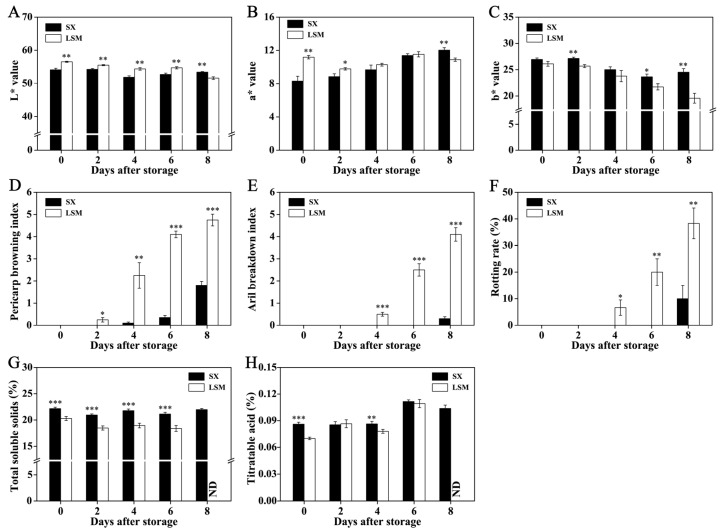
The chromatic values, pericarp browning, aril breakdown, rotting rate, TSS, and TA content of ‘Shixia’ and ‘Luosanmu’ longan fruit during the room-temperature storage. Note: ND: Aril samples were not detected due to serious breakdown. (**A**) *L** value, (**B**) *a** value, (**C**) *b** value, (**D**) Pericarp browning index, (**E**) Aril breakdown index, (**F**) Rotting rate, (**G**) Total soluble solid content, (**H**) Titratable acid content. *: 0.01 < *p* < 0.05; **: 0.001 < *p* < 0.01; ***: *p* < 0.001.

**Figure 2 plants-11-02503-f002:**
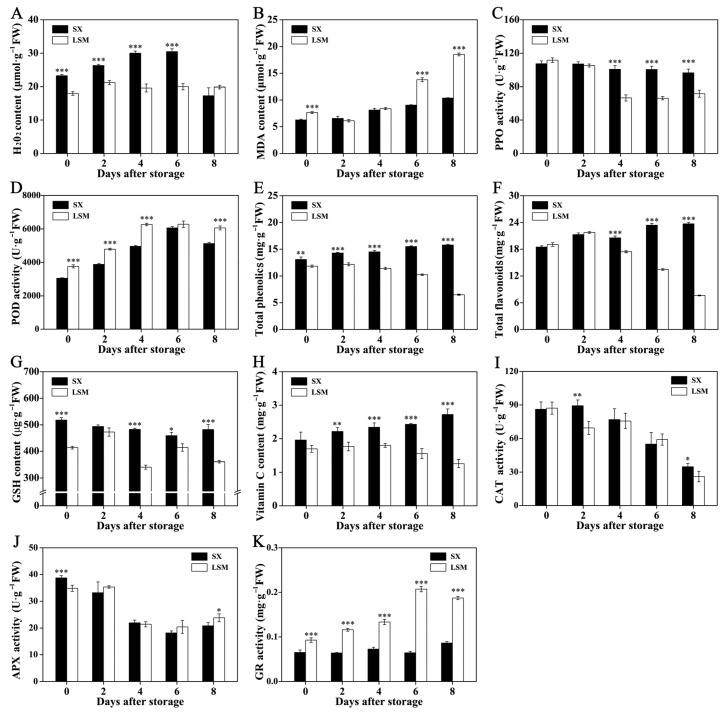
The content of H_2_O_2_ and antioxidants as well as activities of radical-scavenging enzymes in ‘Shixia’ and ‘Luosanmu’ longan pericarp during the room-temperature storage. (**A**) H_2_O_2_ content, (**B**) MDA content, (**C**) PPO activity, (**D**) POD activity, (**E**) total phenolic content, (**F**) total flavonoid content, (**G**) GSH (reduced glutathione) content, (**H**) vitamin C content, (**I**) CAT (catalase) activity, (**J**) APX (ascorbate peroxidase) activity, (**K**) GR (glutathione reductase) activity. *: 0.01 < *p* < 0.05; **: 0.001 < *p* < 0.01; ***: *p* < 0.001. FW: fresh weight.

**Figure 3 plants-11-02503-f003:**
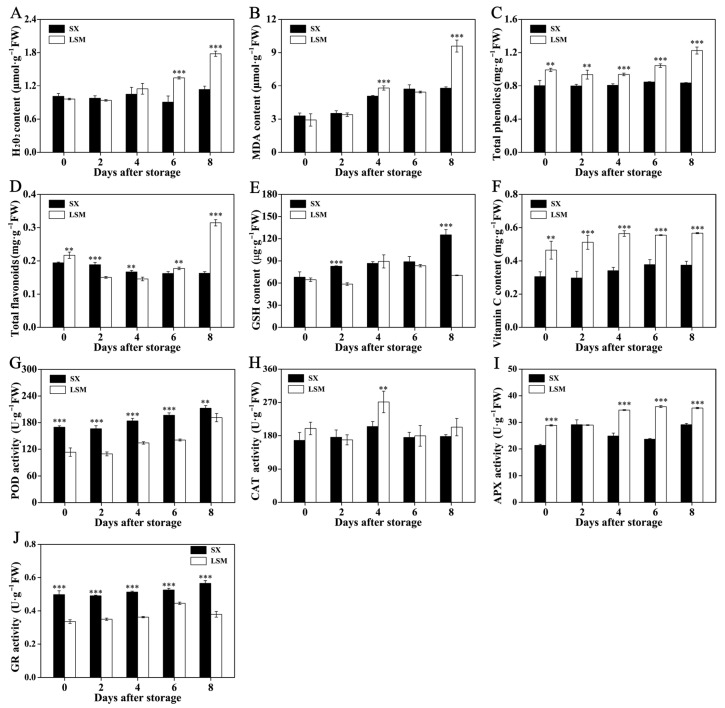
The content of H_2_O_2_ and antioxidants as well as activities of radical-scavenging enzymes of ‘Shixia’ and ‘Luosanmu’ longan aril during the room-temperature storage. (**A**) H_2_O_2_ content, (**B**) MDA content, (**C**) TPC, (**D**) TFC, (**E**) GSH content, (**F**) VC content, (**G**) POD activity, (**H**) CAT activity, (**I**) APX activity, (**J**) GR activity. **: 0.001 < *p* < 0.01; ***: *p* < 0.001. FW: fresh weight.

**Figure 4 plants-11-02503-f004:**
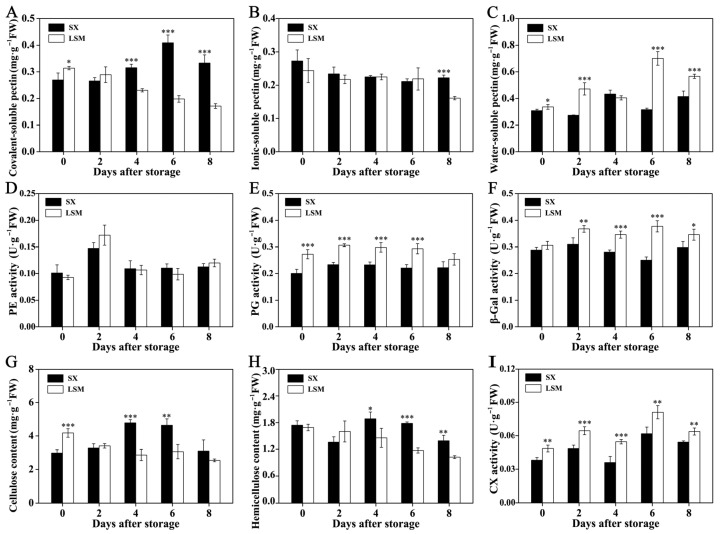
The contents of pectin and cellulose components and activities of the related enzymes of ‘Shixia’ and ‘Luosanmu’ longan aril during the room-temperature storage. (**A**) Covalent-soluble pectin content, (**B**) Ionic-soluble pectin content, (**C**) water-soluble pectin content, (**D**) PE activity, (**E**) PG activity, (**F**) β-Gal activity, (**G**) Cellulose content, (**H**) Hemicellulose content, (**I**) CX activity. *: 0.01 < *p* < 0.05; **: 0.001 < *p* < 0.01; ***: *p* < 0.001. FW: fresh weight.

**Figure 5 plants-11-02503-f005:**
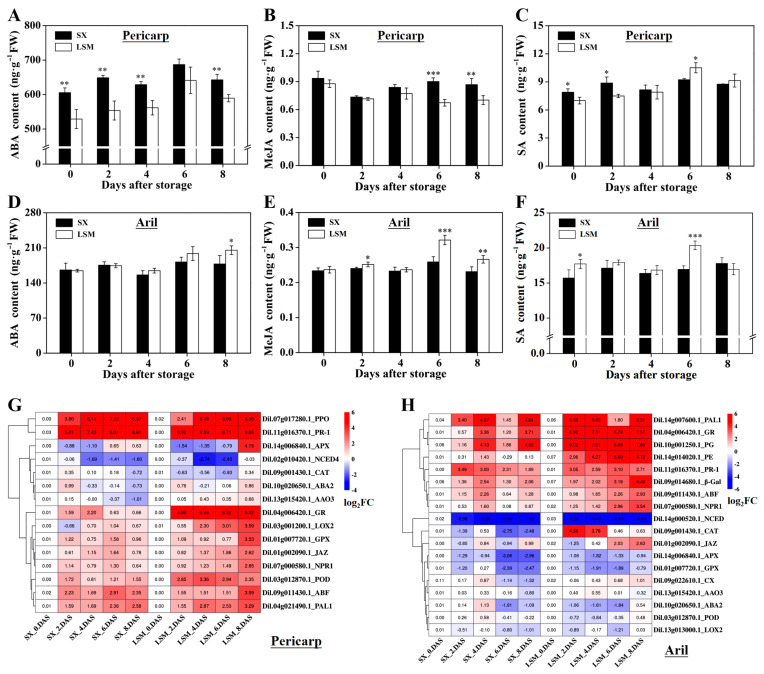
The contents of ABA, JA and SA and the expression of genes related to radical-scavenging, synthesis and signal transduction of hormones in ‘Shixia’ and ‘Luosanmu’ longan fruit. (**A**) ABA, (**B**) JA and (**C**) SA content in pericarp; (**D**) ABA, (**E**) JA, and (**F**) SA content in aril; (**G**) Expression of 15 genes related to radical-scavenging, synthesis and signal transduction of hormones in pericarp; (**H**) Expression of 18 genes related to radical-scavenging, synthesis and signal transduction of hormones in aril. *: 0.01 < *p* < 0.05; **: 0.001 < *p* < 0.01; ***: *p* < 0.001. FC: Fold change (expression of the other samples divided by that of SX_0 DAS). Color gradation: red means up-regulated and blue means down-regulated. FW: fresh weight.

**Figure 6 plants-11-02503-f006:**
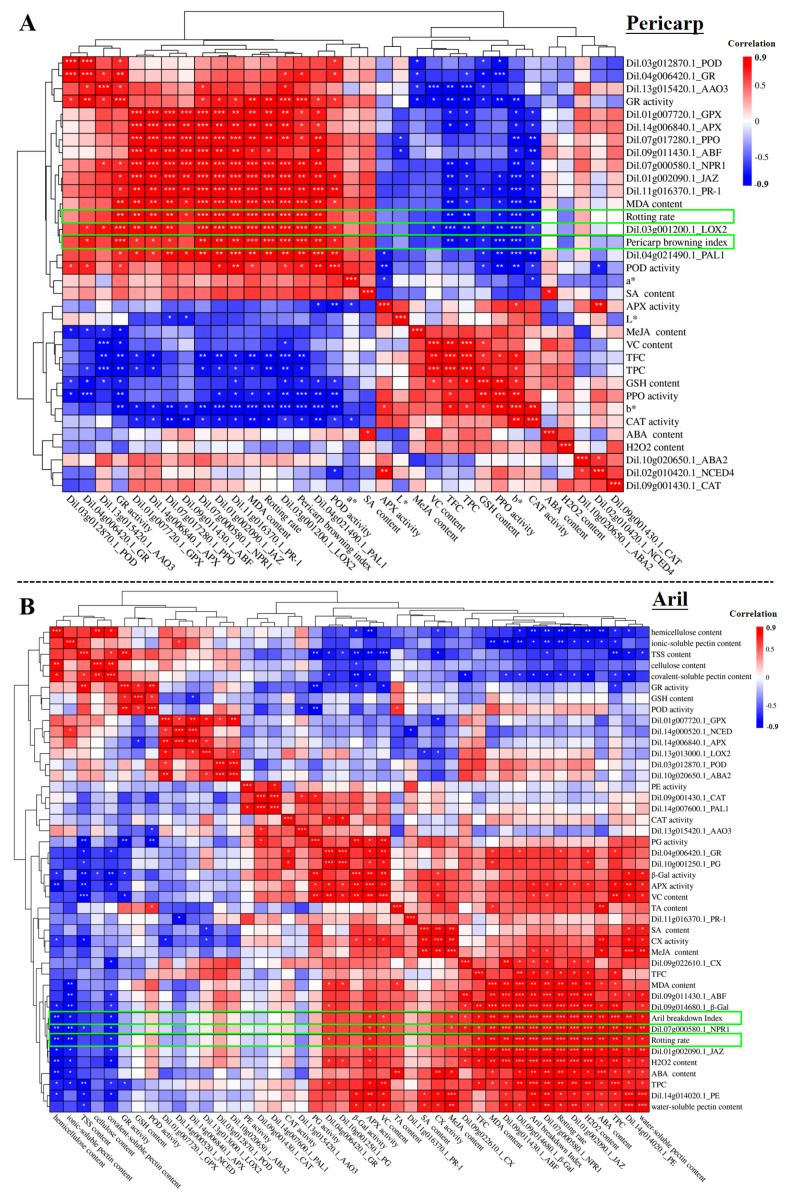
The correlation of physiological parameters, enzyme activities and gene expression with the pericarp browning (**A**) and aril breakdown (**B**). Color gradation: red means positive correlation and blue means negative correlation. The green box marked the correlation of aril breakdown and rooting rate with the other parameters. *: 0.01 < *p* < 0.05; **: 0.001 < *p* < 0.01; ***: *p* < 0.001.

## Data Availability

We did not report any new data in this review.

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
