# Peer review of "A Comprehensive Analysis of Physiologic and Hormone Basis for the Difference in Room-Temperature Storability between ‘Shixia’ and ‘Luosanmu’ Longan Fruits"

_plants, 2022, doi:10.3390/plants11192503_

Round 1

Reviewer 1 Report

Please add a cite in VC determination, pag 12, Line 375-378

In the same page,  change one TFC for TPC, page 12, Line 382.

Author Response

Response to Reviewer 1

Dear Professor or Doctor, 

Thank you very much for your suggestions on revising this paper. We carefully examined the manuscript and gave the answers to the questions point by point as follows:

1) Please add a cite in VC determination, page12, Line 375-378

Answer: We added the citation and corresponding reference [38]. Please see Revised manuscript.doc Page 12, Line 416-417 & Line 738-739.

2) In the same page, change one TFC for TPC, page 12, Line 382.

Answer: The first TFC in Line 382 has been changed to TPC. Please see Revised manuscript.doc Page 12, Line 424. 

Reviewer 2 Report

The manuscript entitled “A Comprehensive analysis of physiologic and hormone basis for the difference in room-temperature storability between “Shixia” and “Luosanmu” Longan Fruits” evidence the behaviour during postharvest storage at room temperature of one tropical and sub-tropical fruit with elevated interest for consumers point of view. Correlation of physiology, biochemistry and hormone content between two cultivars (“Shixia” and “Luosanmu”) of mature longan fruits was revealed. After a carefully reading and review of manuscript I suggest a few improvements, as followed:

In Materials and Methods, regarding the section 4.1 Preparation, treatment and storage of longan fruits, missing information that needs to be added:

a)       the storage days performed for the study, since at Page 11, Line346 “….and stored at room temperature…”

b)      Photographs of the two cultivars of the studied mature longan fruits

Also, indication of the city and country of all equipment’s used should be add, for instance in Page 11, Line 351.

In the section 4.2 Determination of chromatic value, the authors need to add information, mainly the equation of C* and hue values. And the difference of total colour of each fruits cultivars during storage, the authors did not pondered evaluate?

In the section of 4.4 Sampling and determination of total soluble solids (TSS), titratable acid (TA) and vitamin C (VC), Page 12, Line 376, the reference is missing. Which standard was used for VC content?

The procedure for the preparation of samples for the determination of the enzymatic activities is the same/identical, so that there is no repetition of this, it must be initially evidenced/referred before the description of methodology of enzymatic determination.

Author Response

Dear Professor or Doctor,

Thank you very much for your suggestions on revising this paper. We carefully examined the manuscript and gave the answers to the questions point point by point as follows:

In Materials and Methods, regarding the section 4.1 Preparation, treatment and storage of longan fruits, missing information that needs to be added:

a) the storage days performed for the study, since at Page 11, Line346 “….and stored at room temperature…”

Answer: Thanks. The storage days performed for the study was added: After a natural air-dry at room temperature for 30 min, the fruit were packed into trays (about 20 fruits per tray) with 0.01 mm thick polyethylene film and stored at room temperature  for 8 days (RT, 25 ± 1 °C; 85% relative humidity). Please see Revised manuscript.doc Page 12, Line 383.

b) Photographs of the two cultivars of the studied mature longan fruits.

Answer: Thanks. We added the sentence: “Photographs recording changes of appearance and aril breakdown of ‘Shixia’ and ’Luosanmu’ fruits during room- temperature storage were listed in Figure S1.”, Please see Revised manuscript.doc Page 12, Line383-385. The photographs were added in Supplementary Material Figure S1.

c) Also, indication of the city and country of all equipment’s used should be add, for instance in Page 11, Line 351.

Answer: Thanks. We added this information: “The L*, a* and b* values on equatorial plane of each fruit were measured using a color analyzer (NH310, 3nh Technology Co., Ltd., Shenzhen, China).” Please see Revised manuscript.doc Page 12, Line 388-389.

d) In the section 4.2 Determination of chromatic value, the authors need to add information, mainly the equation of C* and hue values. And the difference of total colour of each fruits cultivars during storage, the authors did not pondered evaluate?

Answer: Thanks for your suggestion. We added sentences to elucidate the definition of chromatic value L*, a* and b*: “The value of L* indicates brightness ranging from 0 (black) to 100 (white); The negative value to positive value of a* indicate green to red, respectively; The negative value to positive value of b* indicates blue to yellow, respectively.” Because no significant difference was found between ‘SX’ and ‘LSM’ pericarp, we deleted the assay and equation (calculation) of C* and h° values. Please see Revised manuscript.doc Page 12, Line 387-393.

e) In the section of 4.4 Sampling and determination of total soluble solids (TSS), titratable acid (TA) and vitamin C (VC), Page 12, Line 376, the reference is missing. Which standard was used for VC content?

Answer: We added the citation and corresponding reference [38]. Please see Revised manuscript.doc Page 12, Line 416-417 & Line 738-739.

f) The procedure for the preparation of samples for the determination of the enzymatic activities is the same/identical, so that there is no repetition of this, it must be initially evidenced/referred before the description of methodology of enzymatic determination.

Answer: Thanks for your suggestion. Actually, the determination of the enzymatic activities of each samples at each time point were strictly repeated. So we added this information in the introduction of procedure. Please see Page 14, Line 509: “Each sample at each time point was subjected to three repeated tests.”. Page 15, Line 558-560: “Another supernatant samples were prepared according the above mentioned procedure for determining PG, β-Gal and CX activity, respectively. Each sample at each time point was subjected to three repeated tests. ”.

Reviewer 3 Report

The manuscript entitled "A Comprehensive Analysis of Physiologic and Hormone Basis for the Difference in Room-Temperature Storability between ‘Shixia’ and ‘Luosanmu’ Longan Fruits" by Long et al., was reviewed for publication in Plants (plants-1881521). While the manuscript has a large quantity of potentially useful data, the organisation and writing are very poor, making the manuscript very difficult to read, follow and comprehend. Some general comments to improve the manuscript. There are lot of abbreviations (e.g. CAT, POD, APX, VC, etc…..) that need to be defined the first time used in the text, in addition to the Figure legends and not just in the methods and materials. The way the manuscript is currently written, the reader has to refer constantly to the methods and materials to understand, the Figures are not stand alone to be understandable, and in addition, understanding the main text is difficult. The Results and Discussion sections are hard to read, switching from results to discussion is very difficult to follow, and there is a fair amount of speculation. I would suggest to first describe the results in each section, then finish with the discussion, and follow that pattern throughout the manuscript. Or as an alternative, separate into distinct Results and Discussion sections.  

Line 50, the term “aril” should be defined, because this is particular feature of the Longan fruit.

Lilne 65, does not make sense

Lines 76-78, sentence hard understand

Lines 88-96, sentence long and hard understand, section difficult to read

Lines 106-111, sentence belongs more at the end of the Introduction

Figure 1, Lines 112 -115, Define L*, a* and b* Explain what these values represent.

Legend Figure 1, “quality changes” is very vague, please be more precise. What exactly is being measured?

Lines 148-150, this seems to be speculation

Lines 171-172, this statement is like a conclusion, even before the results are described.

Lines 226-233, these phrases are like a mini introduction/discussion. Start with results and finish with discussion

Figure 5 legend, Line 291, red means increase, blue means decrease? Need to clarify.

Figure 6, red means significant, blue means insignificant? Need to clarify.

The final paragraph is just a long repeat of results, without any integration or the large amount of data. What about the difference in timing observed between the onset of pericarp browning (2-4 days) or aril breakdown and rotting (4 days), and the hormone concentrations etc... It seems like the timing of the physiological aspects (browning, breakdown and rotting) do not match well with that of some of the hormones (e.g. JA at 6 days in both pericarp and aril). It would be helpful to have a final figure that integrates all these results into a scheme that could summarize the most important aspects.

Line 509, CX is defined as cellulose, but I think you mean cellulase. Could be interpreted like this, but need to describe the results first.

Author Response

Dear Professor or Doctor,

Thank you very much for your suggestions on revising this paper. We carefully examined the manuscript and gave the answers to the questions point by point as follows:

The manuscript entitled “A Comprehensive Analysis of Physiologic and Hormone Basis for the Difference in Room-Temperature Storability between ‘Shixia’ and ‘Luosanmu’ Longan Fruits” by Long et al., was reviewed for publication in Plants (plants-1881521). While the manuscript has a large quantity of potentially useful data, the organisation and writing are very poor, making the manuscript very difficult to read, follow and comprehend. Some general comments to improve the manuscript.

1) There are lot of abbreviations (e.g. CAT, POD, APX, VC, etc…..) that need to be defined the first time used in the text, in addition to the Figure legends and not just in the methods and materials.

Answer: We carefully examined these mistakes. See Revised manuscript.doc:

Page 2, Line 55: POD and PPO was firstly explained in the Introduction part: peroxidase (POD), polyphenol oxidase (PPO);

Page 4, Line 126: faster decrease of TSS →faster decrease of soluble solid;

Page 3, Line 107: 0 DAS →0 days after storage (DAS);

Page 3, Line 116: Lower TSS →Lower total soluble solid (TSS);

Page 3, Line 117: Lower TA →lower titratable acid (TA);

Page 4, Line 134: MDA →malondialdehyde (MDA);

Page 4, Line 147-148: GSH →GSH (reduced glutathione);

Page 4, Line 148: VC →vitamin C (VC);

Page 4, Line 149: CAT →catalase (CAT);

Page 4, Line 149-150: APX →ascorbate peroxidase (APX);

Page 4, Line 153: GR →glutathione reductase (GR);

Page 4, Line 141: TPC →total phenolic content (TPC);

Page 4, Line 141: TFC →total flavonoid content (TFC);

Page 6, Line 208: PG →polygalacturonase (PG);

Page 6, Line 209: β-galactosidase (β-Gal);

Page 6, Line 210: PE →pectinesterase (PE);

Page 6, Line 214: CX →cellulose (CX);

Page 8, Line 250-251: NCED1, ABA2 and AAO3 NCED1, nine-cis-epoxycarotenoid dioxygenase; ABA2, xanthoxin dehydrogenase; AAO3, abscisic aldehyde oxidase 3;

Page 8, Line 253: LOX2 LOX2, Lipoxygenase 2;

Page 8, Line 254: PAL1PAL1, phenylalnine ammonialyase 1;

Page 8, Line 255-256: ABF, JAZ and NPR1 ABF, ABA responsive element binding factor; JAZ, jasmonate ZIM domain-containing protein; NPR1, non-expressor of pathogenesis-related gene 1;

Page 8, Line 259: PR-1 PR-1(pathogenesis-related protein 1);

Page 8, Line 261: GPX →GPX-glutathione peroxidase.

2) The way the manuscript is currently written, the reader has to refer constantly to the methods and materials to understand, the Figures are not stand alone to be understandable, and in addition, understanding the main text is difficult.

Answer: Thanks for your suggestions. We carefully examined these mistakes. We added the full name of the abbreviations and revised the Figures legends and presentation of results in the main text. Please see Revised manuscript.doc.

3) The Results and Discussion sections are hard to read, switching from results to discussion is very difficult to follow, and there is a fair amount of speculation. I would suggest to first describe the results in each section, then finish with the discussion, and follow that pattern throughout the manuscript. Or as an alternative, separate into distinct Results and Discussion sections.

Answer: Thanks for your suggestions. We carefully examined these mistakes. We rewrote the Part 2. Results and Discussion: described the results in each section, then finish with the discussion. Please see Revised manuscript.doc.

4) Line 50, the term “aril” should be defined, because this is particular feature of the Longan fruit.

Answer: We explained the term “aril”: Please see Revised manuscript.doc Page 2, Line 49: “edible aril (the hypertrophic tissue attached around the seed coat of longan)”.

5) Line 65, does not make sense.

Answer: Thanks for your advise. In this paragraph, we reviewed the strategies including chemical, physical, and biological treatments to prevent longan fruit from postharvest loss. We also discussed the limits and potential safety hazard of fumigation with SO2 (or other sulfur treatments) and traditional fungicide applied in longan production. So, this paragraph and the discussion about SO2 fumigation was necessary and should be remained.

6) Lines 76-78, sentence hard understand.

Answer: Thanks for your suggestion. We revised this sentence as follows: “Plant hormones act as important factors regulating senescence and responding to biotic and abiotic stresses. Among them, salicylic acid (SA) treatment was reported to suppresses Phomopsis longanae Chi-induced disease development of postharvest longan fruit [9]. However, their behavior, roles and signal transduction in postharvest deterioration of longan fruit were still rarely investigated.” Please see Revised manuscript.doc Page 2, Line 76-80.

7) Lines 88-96, sentence long and hard understand, section difficult to read.

Answer: Thanks for your suggestion. We have modified this paragraph to make it easier to understand. Please see Revised manuscript.doc Page 2, Line 90-96: “ Although plant hormones’ effects on the storability of longan (SA: [9,26,27]) and litchi fruit (abscisic acid, ABA [28]; gibberellic acid, GA3 [28]; melatonin [29]; methyl jasmonate, MeJA [30]) had been reported increasingly in the recent years, their roles in postharvest storage of longan fruit and their possible relationship with the storability differences among cultivars have not been studied. Herein, we systematically compared the biochemistry, enzyme activities and expressions of key genes related to pericarp browning and aril breakdown in ‘Shixia’ to those in ‘Luosanmu’ longan. ”.

8) Lines 106-111,sentence belongs more at the end of the Introduction.

Answer: Thanks for your suggestion. These sentences were integrated into the discussion in the end of this part (part 2.1). Please see Revised manuscript.doc Page 4, Lines 124-130: “In our previous results, it was shown that the ‘Shixia’ fruit had the highest storability among the fruits of 14 cultivars, while ‘Luosanmu’ fruit had a poor performance: higher mass loss rate at 25 ℃, faster decrease of soluble solid, more quick development of pericarp browning and aril breakdown at 4 ℃ [25]. In this work, the above results indicated that the quality of ‘Luosanmu’ longan fruit deteriorated faster than ‘Shixia’ longan fruit during the room temperature storage (Figure1; Figure S1). These results were in accordance with the previous report [25].”

9) Figure 1, Lines 112-115, Define L*, a*and b* Explain what these values represent.

Answer: Thanks for your suggestion. We added the definition about these three chromatic values. Please see Revised manuscript.doc Page 12, Lines 387-392: The value of L* indicates brightness ranging from 0 (black) to 100 (white); The negative value to positive value of a* indicates green to red, respectively; The negative value to positive value of b* indicates blue to yellow, respectively.

Also, we explain what these values represent in the main text. Please see Revised manuscript.doc Page 3, Lines 106-110: Results of chromatic values showed that higher L* (brightness) was found on ‘Luosanmu’ longan pericarp at 0 days after storage (DAS) to 6 DAS and higher a* value (more red) was observed on ‘Luosanmu’ longan pericarp at 0 DAS to 2 DAS (Figure 1A & 1B). No significant difference of b* value (blue to yellow) was found between ‘Shixia’ and ‘Luosanmu’ longan pericarp at harvest (Figure 1C).

10) Legend Figure 1, “quality changes” is very vague, please be more precise. What exactly is being measured?

Answer: Thanks for your suggestion. We rewrote the Legend of Figure 1: “The chromatic values, pericarp browning, aril breakdown, rotting rate, TSS, and TA content of ‘Shixia’ and ‘Luosanmu’ longan fruits during the room-temperature storage.” Please see Page 3, Line 120-121.

11) Lines 148-150, this seems to be speculation.

Answer: Thanks for your suggestion. We rewrote these sentences and this speculation was integrated into the discussion in the end of this part (part 2.2). Please see Revised manuscript.doc Page 5, Line 164-165: Thus, the higher POD activity might be contributed to mediate the participation of H2O2 in lignin polymerization in the ‘Luosanmu’ longan pericarp.

12) Lines 171-172, this statement is like a conclusion, even before the results are described.

Answer: Thanks for your suggestion. These reviewed viewpoints in Line 171-175 was rewritten and integrated into the discussion in the end of this part (part 2.3). Please see Revised manuscript.doc Page 2, Line 186-194.

13) Lines 226-233,these phrases are like a mini introduction/discussion. Start with results and finish with discussion.

Answer: Thanks for your suggestion. These reviewed viewpoint in Line 226-233 was rewritten and integrated into the discussion in the end of this part (part 2.5). Please see Revised manuscript.doc Page 2, Line 278-286: “ABA and ETH are usually considered as two most important regulatory factors of maturity and senescence in climacteric and non-climacteric fruits [33,34]. Especially, ABA was proved to play a vital role in the ripening and postharvest senescence of non-climacteric fruits such as grape [35], blueberry (Vaccinium corymbosum L.) [36], strawberry [37], and citrus [33]. As an important regulator of the defence system, JA shows various defence responses from pathogen to environment stresses [34]. As another inevitable member of the defence system, SA usually enhances environmental resistance at low concentration, but it may induce cell necrocytosis at high concentration [34]. ”.

14) Figure 5 legend, Line 291, red means increase, blue means decrease? Need to clarify.

Answer: Thanks for your suggestion. We added the explanation about the heatmap (Figure 5g-H) in the Figure 5 legend: “FC: Fold change (expression of the other samples divided by that of SX_0 DAS). Color gradation: red means up-regulated and blue means down-regulated.” Please see Revised manuscript.doc Page 9, Line 307-308.

15) Figure 6, red means significant, blue means insignificant? Need to clarify.

Answer: Thanks for your suggestion. We added the explanation about the heatmap in the Figure 6 legend: “Color gradation: red means positive correlation and blue means negative correlation. The green box marked the correlation of aril breakdown and rooting rate with the other parameters.” Please see Revised manuscript.doc Page 11, Line 372-373.

Reviewer 4 Report

Dear Editor, the ms was well prepared and discussed. However, a minor revision is required before it can be considered for publication in Plants. All my comments are in the attached file.

Author Response

Dear Professor or Doctor,

Thank you very much for your suggestions on revising this paper. We carefully examined the manuscript and gave the answers to the questions point by point as follows:

1) Line 25, the term “SA”,what does it mean ? In the same page, change the longan for longan fruit.

Answer: Thanks for your suggestions. We rewrote the first sentence of abstract. Please see Revised manuscript.doc Page 1, Line 23-25: “Although the effects of phytohormone (mainly salicylic acid) on the storability of longan fruits had been reported, the relationship between postharvest hormones’ variation and signal transduction and the storability remains unexplored.”

2) Line 28, the “storability”should provide information about the storage conditions.

Answer: Thanks for your suggestions. We rewrote this sentence, please see Revised manuscript.doc Page 1, Line 25-28: “The basis of physiology, biochemistry, hormone content and signalling for the storability difference at room-temperature between ‘Shixia’ and ‘Luosanmu’ longan fruit were revealed.” We also listed the detailed parameters about the storage. Please see Revised manuscript.doc Page 12, Line 382-383: “stored at room temperature for 8 days (RT, 25 ± 1 °C; 85% relative humidity)”. 

3) Line 29, The correct is “fruit”instead of “fruits”. Fruit is a collective noun taking a singular verb. The plural fruits is used in talking about different types of fruit: oranges, mangoes and other fruits. The whole manuscript must be revised !

Answer: Thanks for your suggestions. We carefully examined these mistakes and revised it throughout the whole manuscript. Please see Revised manuscript.doc Page 1, Line 24; page 2 Line 50, 53, 54; et ...

4) Line 31, change the “Vitamin”for “vitamin”.

Answer: Thanks for your suggestion. Page 1, Line 29, the “Vitamin” has been changed to “vitamin”.

5) Line 35, change the “ABA”for “abscisic acid (ABA), “MeJA” for “methyl jasmonate (MeJA)” 

Answer: Thanks for your suggestions. In the abstract, Line 29, the “ABA” has been changed to “abscisic acid, “MeJA” has been changed to “methyl jasmonate”; Page 2, Line 91-92: the “ABA” has been changed to “abscisic acid (ABA); “MeJA” has been changed to “methyl jasmonate (MeJA)”.

6) Line 79, change the “The previous studies showed”for “In previous studies it was shown” 

Answer: Thanks for your suggestion. Page 2, Line 81, the “The previous studies showed” has been changed to “In previous studies, it was shown that”.

7) Line 86, change the “salicylic acid”for “salicylic acid (SA) ”.

Answer: Thanks for your suggestion. Page 2, Line 77, the “salicylic acid” has been changed to “salicylic acid (SA) ”.

8) Line 89, change the “GA3”for “gibberellic acid (GA3)”

Answer: Thanks for your suggestion. Page 2, Line 91, the “GA3” has been changed to “gibberellic acid (GA3)”.

9) Line 106, change the “showed”for “had”

Answer: Thanks for your suggestion. Page 3, Line 124-125, the “showed” has been changed to “had” .

10) Line 107, change the “varieties”for “cultivars”

Answer: Thanks for your suggestion. Page 3, Line 125, the “varieties” has been changed to “cultivars”.

11) Line 287-288, figure 5, I believe Figures G and Hshould be in a different Figure, provided that the way it is, it is almost impossible to read the letters. These images must be enlarged !

Answer: Thanks for your suggestion. In order to compare the differences of ABA, MeJA and SA contents between ‘Shixia’ and ‘Luosanmu’, and find the possible correlation of the expressions of related genes with the hormones’ content, Figures 5G and 5H were placed together with Figure 5A-F. In addition, we enlarged the Figure 5 and Figure 6 to be more clear.

12) Line 288, the title of Figure 5, the species and cultivars must be added.

Answer: Thanks for your suggestion. Line 288, the title of Figure 5 was revised as: “The contents of ABA, JA and SA and the expression of genes related to radicals-scavenging, synthesis and signal transduction of hormones in ‘Shixia’ and ‘Luosanmu’ longan fruit.” Please see Page 9, Line 303-304.

Round 2

Reviewer 3 Report

I made comments directly on the uploaded pdf

Author Response

Response to Reviewer 3

Thank you very much for your suggestions on revising this paper. We carefully examined the manuscript and gave the answers to the questions point by point as follows:

(1) Abstract: “revealed” → “examined”. examined would be more appropriate here. Answer: Thanks for your suggestion. We replaced “Revealed” by “examined”.

(2) Introduction:

(a) The phrase in the first paragraph: “ Except for the induction by reactive oxygen species, the destroying of cellular compartmentation and integrity of cell membrane, which was the most important factors promoting postharvest longan pericarp browning, might also be induced by water loss [8], pathogen infection [9], and other stresses.” was unclear.

Answer: Thanks for your suggestion. We rewrote this sentence: “Except being induced by reactive oxygen species, postharvest longan pericarp browning was also promoted by the destroying of cellular compartmentation and integrity of cell membrane, which was usually resulted from water loss [8], pathogen infection [9], and other stresses.”

(b) The second paragraph, Line 4: “were”→ are still used? or no longer used?

Answer: Thanks for your suggestion. We replaced “were” by “are”.

(c) The third paragraph, Line 2: “This ” → “These”.

Answer: Thanks for your suggestion. We replaced “This” by “These”.

(d) In the last paragraph of the Introduction, it is stated “In previous studies, it was shown that the occurrence and development of aril breakdown was different between ‘Fuyan’ and ‘Dongbi’ longan fruit [23,24].” Then at the end of the paragraph, it is stated that “Herein, we systematically compared the biochemistry, enzyme activities and expressions of key genes related to pericarp browning and aril breakdown in ‘Shixia’ to those in ‘Luosanmu’ longan.” So it is confusing and unclear whether ‘Fuyan’ and ‘Dongbi’ are the same as ‘Shixia’ to those in ‘Luosanmu’ ?

Answer: Actually, in previous studies, it was shown that faster development of aril breakdown and worse storability was observed in ‘Fuyan’ longan fruit when compared to ‘Dongbi’ longan fruit stored at room-temperature (25 ± 1 °C) [23,24]. Similar result was found in our previous work that the difference in low-temperature (4 °C) storability among 14 longan cultivars was compared. More importantly, ‘Shixia’ longan fruit and ‘Luosanmu’ longan fruit showed the best storability and the worst storability, respectively [25]. The difference in storability between ‘Luosanmu’ and ‘Shixia’ was greater than the in storability between ‘Fuyan’ and ‘Dongbi’ longan fruit. However, the reason for the great difference in storability between ‘Shixia’ and ‘Luosanmu’ longan fruit remains to be studied. Thus, we conducted a comprehensive analysis of physiologic and hormone basis for the difference in room-temperature storability between ‘Shixia’ and ‘Luosanmu’ longan fruits.

We rewrote this paragraph: “In previous studies, it was shown that faster development of aril breakdown and worse storability was observed in ‘Fuyan’ longan fruit when compared to ‘Dongbi’ longan fruit stored at room-temperature (25 ± 1 °C) [23,24]. Similar result was found in our previous work that the difference in low-temperature (4 °C) storability among 14 longan cultivars was compared. It was also shown that postharvest deterioration including mass loss, pericarp browning and aril breakdown varied among different cultivars. More importantly, ‘Shixia’ longan fruit and ‘Luosanmu’ longan fruit showed the best storability and the worst storability, respectively [25]. However, the reason for the great difference in storability between ‘Shixia’ and ‘Luosanmu’ longan fruit remains to be studied. On the other hand, although plant hormones’ effects on the storability of longan (SA: [9,26,27]) and litchi fruit (abscisic acid, ABA [28]; gibberellic acid, GA3 [28]; melatonin [29]; methyl jasmonate, MeJA [30]) have been reported increasingly in the recent years, their roles in postharvest storage of longan fruit and their possible relationship with the storability differences among cultivars have not been studied. Herein, in order to uncover the physiologic and hormone basis for the difference in room-temperature storability between ‘Shixia’ and ‘Luosanmu’ longan fruits, we systematically compared the biochemistry, enzyme activities and expression of key genes related to pericarp browning and aril breakdown in ‘Shixia’ to those in ‘Luosanmu’ longan. The contents of ABA, MeJA and SA as well as the expression of genes related to their biosynthesis and signal transduction were also detected in the pericarp and aril of ‘Shixia’ and ‘Luosanmu’ during room-temperature storage. Through a correlation analysis, we screened out the key gene, enzyme and phytohormone, which were highly associated with the pericarp browning and aril breakdown. These results are expected to provide new insights into the possible mechanism regulating storability differences among cultivars of longan.”.

(3) Results and Discussion:

(a) Part 2.1, “Results of chromatic values” → “Results of chromatic analysis”

Answer: Thanks for your suggestion. We replaced “values” by “analysis”.

(b) Part 2.1, “The pericarp browning showed a consistent trend with this”, what is “this”.

Answer: Thanks for your suggestion. We revised this sentence: “The browning of inner pericarp showed a consistent trend with the deterioration of appearance (Figure 1D; Figure S1).”.

(c) Part 2.2, “radicals-scavenging” → “radical-scavenging”.

Answer: Thanks for your suggestion. We replaced “radicals” by “radical” throughout the whole text.

(d) Figure 2, legend, give the full name of GSH. Also give the full namesof TPC and TFC when they were firstly used in the text.

Answer: Thanks for your suggestion. We gave the full names of GSH, TPC and TFC when they were firstly used in the text in Part 2.2.

(e) Part 2.3, 2th paragraph Line 3, “a increased” → “ an increase”.

Answer: Thanks for your suggestion. We replaced “a increased” by “an increase of”.

(f) Part 2.3, 2th paragraph Line 6, “this” → “ the current work”.

Answer: Thanks for your suggestion. We replaced “this work” by “the current work”.

(g) Part 2.4, 1th paragraph Line 11-13. “It was interesting that parallelly higher cellulose (CX) activity was observed in ‘Luosanmu’ longan aril throughout the whole storage (Figure 4I).” →“It was interesting that higher cellulose (CX) activity was also observed in ‘Luosanmu’ longan aril throughout the whole storage (Figure 4I).”

Answer: Thanks for your suggestion. We revised this sentence: “It was interesting that higher cellulose (CX) activity was also observed in ‘Luosanmu’ longan aril throughout the whole storage (Figure 4I).

(h) Part 2.5, 1th paragraph Line 11-13. “parallelly” → “consistently”.

Answer: Thanks for your suggestion. We replaced “parallelly” by “consistently”.

(i) Part 2.5, 2th paragraph Line 3, 12, 14: “expressions” → “expression”.

Answer: Thanks for your suggestion. We replaced “expressions” by “expression”.

(j) Part 2.5, 2th paragraph Line 16:“radicals-scavenging”→“radical-scavenging”

Answer: Thanks for your suggestion. We replaced “radicals” by “radical” throughout the whole text.
